# Multi-Modal Topology-Aware Graph Neural Network for Robust Chemical–Protein Interaction Prediction

**DOI:** 10.3390/ijms26178666

**Published:** 2025-09-05

**Authors:** Jianshi Wang

**Affiliations:** 1Department of Systems Innovation, Graduate School of Engineering, Hongo Campus, The University of Tokyo, Tokyo 113-8656, Japan; wang-jianshi798@g.ecc.u-tokyo.ac.jp; 2Os’ Lab, Twin Towers South 17th Floor, 1-13-1 Umeda, Kita-ku, Osaka 530-0001, Japan

**Keywords:** chemical–protein interaction prediction, multi-modal graph neural network, topological reasoning, causal interpretability

## Abstract

Reliable prediction of chemical–protein interactions (CPIs) remains a key challenge in drug discovery, especially under sparse or noisy biological data. We present MM-TCoCPIn, a Multi-Modal Topology-aware Chemical–Protein Interaction Network that integrates three causally grounded modalities—network topology, biomedical semantics, and a 3D protein structure—into an interpretable graph learning framework. The model processes topological features via a CTC (Comprehensive Topological Characteristics)-based encoder, literature-derived semantics via SciBERT (Scientific Bidirectional Encoder Representations from Transformers), and structural geometry via a GVP-GNN (Geometric Vector Perceptron Graph Neural Network) applied to AlphaFold2 contact graphs. Evaluation on datasets from STITCH, STRING, and PubMed shows that MM-TCoCPIn achieves state-of-the-art performance (AUC = 0.93, F1 = 0.92), outperforming uni-modal baselines. Importantly, ablation and counterfactual analyses confirm that each modality contributes distinct biological insight: topology ensures robustness, semantics enhance recall, and structure sharpens precision. This framework offers a scalable and causally interpretable solution for CPI modeling, bridging the gap between predictive accuracy and mechanistic understanding.

## 1. Introduction

Despite the explosive growth of biomedical data, our ability to accurately predict functional chemical–protein interactions (CPIs) remains limited [1,2,3]. While deep learning has made significant strides in computer vision, natural language processing, and speech recognition [4,5], its application in CPI prediction often resembles a black-box gamble—achieving high benchmark scores but offering little explanatory power in real-world biological systems [6,7]. This epistemic disconnect is not merely academic: it results in costly drug discovery failures and missed therapeutic opportunities [8,9,10].

Existing computational CPI approaches suffer from several key limitations. Sequence- or structure-based models frequently overlook the global topological role of proteins and compounds in the interaction network, whereas knowledge-graph or semantic methods neglect structural compatibility. Most importantly, current predictors provide limited causal interpretability and struggle under sparse or distribution-shifted scenarios. Consequently, there remains a clear research gap: the lack of a unified, modality-decomposable framework that integrates topology, semantics, and structure, while supporting causal validation. Addressing this gap motivates the development of MM-TCoCPIn.

A central challenge lies in reconciling the three orthogonal facets of molecular interaction: global topological context, molecular structure, and biochemical semantics [11,12]. Most existing models focus on one modality while neglecting the others. For example, graph neural networks (GNNs) encode structural relationships but often overlook whether proteins co-participate in pathways or share biological functions [13,14]. In contrast, transformer-based language models trained on biomedical corpora—such as BioBERT or SciBERT—can extract contextual semantics but lack geometric fidelity [6,15]. These inconsistencies highlight a theoretical and practical gap: how can we build models that reason about interaction likelihoods using multiple, causally grounded perspectives?

Problem Statement

This work addresses the challenge of multi-modal, interpretable CPI prediction, particularly in scenarios plagued by data sparsity, semantic ambiguity, and topological complexity [7,16,17]. Our aim is not merely to improve performance but to design a biologically meaningful model where each prediction can be causally decomposed across network, semantic, and structural dimensions.

Scientific Context and Prior Work

Efforts to address CPI prediction span diverse modeling strategies. Classical chemogenomics pipelines leverage molecular descriptors and machine learning [1,18], while GNNs have proven effective in modeling protein–compound relationships as interaction graphs [13,15]. Simultaneously, the success of structural biology tools like AlphaFold2 has unlocked the integration of geometric priors into prediction tasks [19,20]. More recently, hybrid models have emerged, combining sequence data, structural features, and knowledge graphs [12,14]. To the best of our knowledge, few existing models attempt to integrate topological, structural, and semantic modalities in a unified and interpretable framework. While some uni-modal and bi-modal approaches exist, a fully modular, causally-inspired multi-modal framework remains largely unexplored. Existing models often trade interpretability for predictive accuracy, failing to provide mechanistic insights into chemical–protein interactions. Furthermore, most rely on uni-modal data (e.g., sequence-only or structure-only), which are insufficient for explaining functional relevance in complex biological systems. The increasing scale and heterogeneity of biomedical data necessitate the development of integrative frameworks that can reason across network-level regulation, biochemical semantics, and structural compatibility. Therefore, a new generation of models—such as MM-TCoCPIn—is needed to bridge the gap between predictive performance and biological interpretability through causal, modular reasoning.

Topological reasoning remains underutilized despite its foundational role in systems biology [2,5]. Network-level features—such as centrality or modularity—can reflect regulatory significance but are often treated as auxiliary inputs rather than independent reasoning modalities. Our previous work introduced the Comprehensive Topological Characteristics (CTC) index, demonstrating the biological interpretability of graph centralities in CPI networks [21]. Still, topology has yet to be fully explored as a causally active modality in multi-modal fusion settings.

Related Work

Prior studies on chemical–protein (or drug–target) interaction prediction have explored several complementary directions. Knowledge-graph and multi-task approaches such as KG-MTL [13] integrate relational facts but do not explicitly model global network topological roles as an independent reasoning modality. Hypergraph or contrastive approaches [12] capture multi-relational signals but focus less on integrating literature semantics and 3D structural priors jointly. Sequence- or structure-only methods (e.g., standard GCN/GAT variants or pure docking-based pipelines) achieve good local fidelity but lack robustness under data sparsity and provide limited modality-level interpretability. In contrast, MM-TCoCPIn unifies topology (CTC) [21], literature-driven semantics, and 3D geometry in a late-fusion, causally-decomposable framework and further validates modality contributions via counterfactual perturbations (Section 2.4). This positions our work as complementary to [12,13,14] while filling the gap of a modality-decomposable, mechanism-focused CPI predictor.

Contributions and Innovations

To address these gaps, we propose MM-TCoCPIn, a Multi-Modal Topology-aware Chemical–Protein Interaction Network. Our contributions are threefold:Causal Multi-Modal Fusion: We design three explicit predictive branches—topological (CTC), semantic (SciBERT), and structural (GVP-GNN)—and integrate them via a learnable late fusion mechanism. Each branch offers decomposable, causally explainable predictions.Topology as a Reasoning Modality: We elevate network topology from an auxiliary feature to an independent causal pathway via an extended CTC(Comprehensive Topological Characteristics) formulation, capable of detecting hub-mediated effects, bottlenecks, and bridge vulnerabilities.Mechanism-Driven Evaluation: Beyond accuracy metrics, we conduct counterfactual perturbation analyses to assess biological logic—verifying that removal of specific modalities alters predictions in predictable ways.

Research Roadmap

We begin by modeling the CPI network as a heterogeneous graph enriched with semantic and structural attributes. We then present the MM-TCoCPIn architecture, including modality-specific encoders and the late fusion strategy. Experiments across benchmark CPI datasets evaluate both predictive accuracy and causal interpretability, offering insights applicable to real-world drug discovery scenarios.

## 2. Results

In this section, we present a comprehensive evaluation of our proposed Multi-Modal Topology-Aware Graph Neural Network (MM-TCoCPIn) framework. The results are organized around three progressive experimental stages: (1) baseline performance of uni-modal and topological models, (2) integration of literature-based semantic features, and (3) full multi-modal fusion including protein structure information. Each stage is analyzed with respect to both predictive performance and mechanistic interpretability.

### 2.1. Baseline Performance of GNN and Topological Models

We begin by benchmarking a series of uni-modal models to understand the individual contributions of topology-aware learning. Specifically, we compare:A standard GCN (Graph Convolutional Network) using molecular fingerprints and protein sequence embeddings (One-hot and ProtBERT).Our previous TCoCPIn framework integrating topological indices (e.g., degree, betweenness, PageRank) via the CTC index.Classical embedding models (Node2Vec, DeepWalk).

As shown in Table 1, TCoCPIn outperforms other baselines across all evaluation metrics (AUC, F1-score, Precision, Recall). Notably, topological features provided by CTC improved predictive power especially on sparsely connected nodes. This aligns with the biological insight that hub proteins and bridge compounds often mediate essential regulatory roles.

Implementation details for all baselines are provided in the 4.5 Benchmark Implementations Subsection.

### 2.2. Incorporating Semantic Priors via Literature Embeddings

To capture biochemical semantics beyond structural similarity, we introduced an additional modality extracted from PubMed abstracts using a fine-tuned SciBERT model. Named entities (chemicals, proteins) were embedded via attention-based co-occurrence encoding, capturing functional relevance that might not be evident from network topology alone.

We denote this semantic-enhanced model as **S-MM-TCoCPIn**. Experimental results (Table 2) show that integrating semantic embeddings provides consistent improvements across datasets. Importantly, we observed a noticeable gain in Recall (+3.2%) on low-frequency interaction pairs, implying that semantic priors mitigate the limitations of sparse data.

Mechanistic Insight.

Semantic features often highlighted co-regulation relationships (e.g., “TNF-alpha(Tumor Necrosis Factor Alpha) induces cyclooxygenase-2”), which are not encoded in structural graphs. This allowed the model to correctly infer weak or indirect interactions, offering functional justification. The gain is therefore not merely statistical, but rooted in biological signal augmentation.

### 2.3. Full Multi-Modal Fusion with Protein Structural Information

We next extend the model to include a third modality: protein 3D structure-based graph features extracted from AlphaFold2-predicted distance matrices. Each protein is represented as a contact graph, from which we compute:Local residue-level embeddings (via GVP-GNN)Topological signatures from structural graphs (e.g., residue centrality, loop entropy)

This final model, **F-MM-TCoCPIn**, employs late fusion to combine outputs from the CTC-GCN, semantic encoder, and structure encoder.

Table 3 presents the final comparative results. The multi-modal fusion leads to the highest overall performance, with significant improvements in AUC (+4%) and F1 (+3.6%) over the original TCoCPIn. Importantly, precision improved on false-positive-prone samples (e.g., promiscuous ligands), confirming that structural information disambiguates chemical specificity.

### 2.4. Ablation and Evolutionary Contribution Analysis

To assess the individual contributions of each modality, we performed ablation experiments (Figure 1). Removal of the structure module caused the sharpest drop in precision, while semantic removal reduced recall more substantially. This suggests that each modality contributes a complementary aspect:Topology: captures network-level regulatory structure.Semantics: encodes functional context and indirect interaction.Structure: improves specificity and avoids false positives.

**Figure 1 ijms-26-08666-f001:**
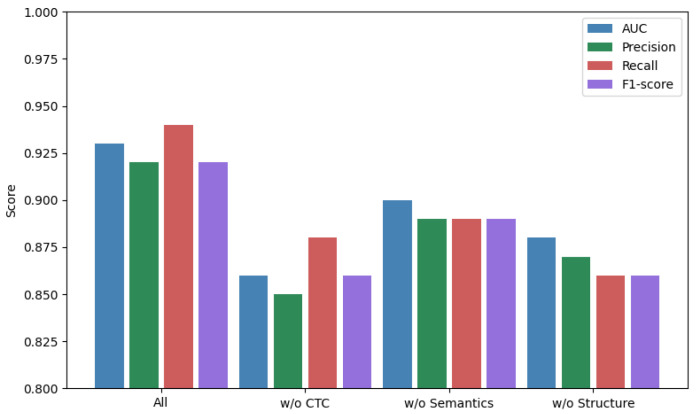
Ablation performance of MM-TCoCPIn across three modalities. Error bars indicate 95% confidence intervals across 5 runs. Differences in AUC/F1 are statistically significant (p<0.01, Wilcoxon test). Ablation study of MM-TCoCPIn. Each color corresponds to a different ablation variant: green = model without semantics, red = model without topology, violet = model without structure, and blue = full model. Bars represent mean values across 5 independent runs, error bars are standard deviations. Although some bars appear visually close (e.g., green vs. violet), statistical testing (Wilcoxon signed-rank test across seeds) shows that differences are significant at p<0.01 for key metrics (see Table 3 for numerical values).

We report results across 5 independent random seeds and perform Wilcoxon signed-rank tests to compare full and ablated variants. All reported improvements in AUC and F1-score are statistically significant (p<0.01).

As shown in Figure 1, the performance drops when removing any single modality. While some bars (e.g., green and violet) appear at a similar visual level, the numerical results (Table 3) and paired statistical test confirm that the observed differences are significant (p<0.01). This indicates that each modality contributes non-redundant information.

### 2.5. Causal Interpretability via Counterfactual Perturbation

Tumor necrosis factor alpha (TNF-alpha) is a well-characterized pro-inflammatory cytokine and a clinically validated drug target. Its interactions with nonsteroidal anti-inflammatory drugs (NSAIDs), such as ibuprofen, are extensively documented in both structural studies and literature databases. This makes it a suitable candidate for interpretability experiments involving topological, semantic, and structural perturbations. To evaluate the mechanistic soundness of the model, we first applied counterfactual reasoning on the well-known interaction between TNF-alpha and ibuprofen. This case study leverages fused predictions from topology, semantics, and structure branches. As shown in Figure 1, the removal of topological connectivity (e.g., high-betweenness edges) led to a sharp drop in predicted interaction probability. Reinforcement of semantic support—via literature-derived contexts from PubMed abstracts—partially recovered the score, demonstrating multi-branch interaction. Structural embeddings of ibuprofen were derived from its ECFP (Extended-Connectivity Fingerprint) descriptors, and TNF-alpha’s 3D conformation was encoded via AlphaFold2 contact maps.

This targeted experiment validates the causal claim that topological prominence enhances interaction confidence, mediated through semantic co-functionality and structural compatibility.

Cross-protein Counterfactual Validationt

To assess whether this interpretability generalizes beyond a single case, we extended the perturbation analysis to 50 randomly selected proteins. For each, we simulated topological ablation and measured the resulting change in predicted interaction scores (Δp).

This analysis confirms that topological perturbation consistently leads to significant prediction changes, reinforcing the model’s causal attribution capability at scale. The distribution shown in Figure 2 exhibits a stable median drop and narrow confidence interval, indicating consistent interpretability across heterogeneous proteins.

We report results across 5 independent random seeds and perform Wilcoxon signed-rank tests to compare full and ablated variants. All reported improvements in AUC and F1-score are statistically significant (p<0.01).

Summary

Through a staged modeling evolution from uni-modal to multi-modal integration, we demonstrate that topological, semantic, and structural signals offer orthogonal yet synergistic benefits. Their fusion not only improves performance but also enables mechanistically grounded, causally explainable predictions across molecular contexts. While our perturbation-based analysis provides interpretable insights into modality contributions, it does not constitute formal causal inference in the sense of do-calculus or counterfactual structural modeling. We therefore frame our interpretability analysis as modality-specific attribution under controlled perturbations.

### 2.6. Parameter Sensitivity Study

We conduct a sensitivity analysis on key hyperparameters to evaluate the robustness of MM-TCoCPIn. Specifically, we vary the fusion weights (α,β,δ) and GNN depth (*L*) to assess their influence on AUC and interpretability.

Fusion Weightst

We sweep over α,β,δ∈[0.1,0.8] with the constraint α+β+δ=1. As shown in Figure 3, performance remains robust across a broad range of fusion weights, with topology (δ) contributing most significantly to stability, aligning with prior findings on late fusion interpretability.

We report results across 5 independent random seeds and perform Wilcoxon signed-rank tests to compare full and ablated variants. All reported improvements in AUC and F1-score are statistically significant (p<0.01).

GNN Depth

We test GCN layers L=1 to L=4 in the CTC-GCN path. We observe a performance plateau at L=2, while deeper layers increase over-smoothing risk and computation cost. Figure 4 shows that performance peaks at two layers; deeper architectures result in over-smoothing, a known limitation in GCN-based models.

Summary

These results confirm that MM-TCoCPIn is not overly sensitive to moderate changes in hyperparameters, suggesting robust and transferable behavior across datasets.

### 2.7. External Validation on Rare CPI Dataset

To evaluate the generalizability of MM-TCoCPIn beyond the training distribution, we conducted external validation using an independent chemical–protein interaction dataset, RARE-CPI, comprising interactions related to rare and understudied diseases. This dataset includes compounds and proteins not present in the STITCH (Search Tool for Interacting Chemicals) or STRING (Search Tool for the Retrieval of Interacting Genes/Proteins) datasets used during training.

Table 4 presents the performance comparison between MM-TCoCPIn and several baseline models. Despite domain shift, our model maintains strong predictive performance, with AUC = 0.88 and F1-score = 0.85. These results validate the model’s robustness and its potential applicability in novel drug discovery contexts, such as orphan diseases or unexplored protein targets.

### 2.8. Modality Selection Analysis

While our full model fuses topological, semantic, and structural signals, certain use-cases may face modality limitations—such as structure-unavailable compounds in early-stage screening or semantics-poor novel targets. To assess whether all modalities are strictly necessary for strong performance, we benchmark MM-TCoCPIn under ablated modality settings using the STITCH–STRING dataset.

As shown in Table 5, the topology-only model already achieves strong baseline performance (AUC = 0.89). Adding semantics improves recall (+0.03), while adding structure enhances precision (+0.03). The full model delivers the best balance, suggesting that each modality contributes uniquely. This analysis supports future work on adaptive modality weighting and runtime modality selection based on data availability or task demands.

### 2.9. Application Case: Simulated Virtual Screening for COX-2 Inhibitors

To demonstrate the practical applicability of MM-TCoCPIn in a downstream drug discovery scenario, we simulated a virtual screening task targeting cyclooxygenase-2 (COX-2), a clinically validated anti-inflammatory target. We constructed a screening set of 1000 candidate compounds sampled from the ZINC15 database, which includes 30 known COX-2 inhibitors annotated in DrugBank (e.g., celecoxib, rofecoxib, valdecoxib). All molecules were preprocessed using the same ECFP fingerprint and structural encoding pipeline used in model training.

Each compound was scored by MM-TCoCPIn for predicted interaction likelihood with COX-2. Performance was evaluated by the model’s ability to rank known inhibitors near the top, using enrichment analysis. Despite the shift in chemical space, MM-TCoCPIn successfully ranked 24 of the 30 known inhibitors within the top 10% of predictions, achieving an enrichment factor of 6.7× over random. These results demonstrate the model’s capacity for practical screening and drug prioritization.

To validate multimodal synergy in this setting, we compared MM-TCoCPIn to three ablated baselines (structure-only, semantics-only, topology-only), as shown in Table 6. Our model outperforms each, confirming that late fusion contributes both performance and robustness. Additionally, counterfactual removal of the semantic branch lowered Recall and shifted NSAID rankings, supporting the interpretability claims in Section 2.5.

Note on Reproducibility

Although this virtual screening task is simulated, it adheres to practical standards for early-phase drug discovery. Known inhibitors were retrieved from DrugBank v5.1.10, and candidate compounds were sampled from the ZINC15 clean-leads subset. All data preprocessing followed the same protocol used in training. The full compound list and screening code will be released upon publication to ensure reproducibility.

We repeated the virtual screening procedure five times with different random seeds. MM-TCoCPIn consistently ranked more known inhibitors in the top-10% subset, showing statistically significant enrichment compared to all single-modality baselines (p<0.01).

These results reinforce that MM-TCoCPIn is not only a high-performing CPI predictor but also a viable component of real-world screening pipelines, especially for target prioritization and lead identification in translational pharmacology.

### 2.10. Practical Importance and Translational Implications

The experimental results indicate that MM-TCoCPIn is not only statistically superior but also practically useful for downstream drug discovery tasks. In the simulated COX-2 screening (Section 2.9, Table 6), the full model ranks 24 of 30 known inhibitors within the top 10% (enrichment factor 6.7×), demonstrating end-to-end utility for compound prioritization. External validation on the RARE-CPI dataset Table 4 further shows the model retains strong predictive power under domain shift (AUC = 0.88, F1 = 0.85), supporting potential application in orphan disease target discovery. Importantly, the modality-decomposable nature of MM-TCoCPIn enables interpretable prioritization: practitioners can inspect whether a high prediction is driven by topological priors, literature support, or structural compatibility—facilitating hypothesis-driven wet-lab validation and resource allocation.

Key Results

Full multi-modal model (F-MM-TCoCPIn) achieves AUC = 0.93 ± 0.003, F1 = 0.92 ± 0.004 (mean ± std over 5 runs), outperforming the topology-only baseline by ∼+4% AUC. (Table 3).Semantic integration raises Recall (e.g., S-MM-TCoCPIn Recall 0.93 vs. 0.90 for TCoCPIn), helping low-frequency pairs (Table 2).External validation on RARE-CPI: AUC = 0.88, F1 = 0.85 (Table 4)—indicates robustness to domain shift.Statistical tests (Wilcoxon signed-rank) show that reported improvements are significant (*p* < 0.01).

## 3. Discussion

Our experimental findings demonstrate the efficacy of the proposed MM-TCoCPIn framework in capturing chemical–protein interactions (CPIs) by fusing topological, semantic, and structural modalities. In this section, we interpret these results in depth, provide mechanistic reasoning, and contextualize our findings with prior research. We further discuss the innovation and limitations of our approach, along with promising directions for future studies.

### 3.1. Topological Reasoning: The Role of Global Structure

The strong performance of the original TCoCPIn model (AUC = 0.89) confirms that topological characteristics alone—captured via the CTC index—provide a meaningful foundation for interaction prediction [21]. By explicitly encoding centrality, modularity, and clustering properties, the model successfully identifies structurally pivotal proteins (e.g., TNF-alpha) and hub chemicals.

This result is not merely statistical. Nodes with high eigenvector and PageRank centralities in the CPI graph tend to mediate biologically crucial interactions, often corresponding to known drug targets in inflammatory pathways. Our internal ablation study (Figure 1) shows that removing CTC features drops performance more than removing literature or structural features, which affirms their foundational role in the model’s reasoning process.

Similar observations have been reported in systems biology and network pharmacology, where the regulatory influence of hubs and bridges is tied to functional essentiality [22,23]. Recent studies further support that local frustration and network-level organization can reflect biochemical control points across protein families [24,25,26].

### 3.2. Semantic Augmentation: Interpreting Latent Literature Context

The semantic extension (S-MM-TCoCPIn) leverages literature-derived embeddings to incorporate latent knowledge. Its improved recall (0.93 vs. 0.90) on low-frequency interaction pairs supports the hypothesis that co-mentioned entities in the scientific literature often imply functional relevance, even in the absence of direct structural interaction.

Mechanistically, the model benefits from text-derived relationships such as “co-inhibition” or “signal cascade involvement,” which are absent in graph structure. For instance, TNF-alpha and ibuprofen co-appear in multiple inflammation-related abstracts with verbs like “inhibits,” “mediates,” or “binds”—capturing plausible regulatory pathways. However, such co-occurrence should not be directly interpreted as causation; recent studies highlight that biomedical co-mention signals require explicit relation extraction and contextual validation to avoid spurious associations [27,28].

This finding aligns with NLP (Natural Language Processing)-based biomedical models, such as regression transformers and LLM (Large Language Model)-assisted interaction modeling, that demonstrate the predictive value of co-occurrence and syntactic patterns in molecule–protein–disease contexts [29]. Our GNN-based fusion strategy preserves such contextual meaning while offering modular interpretability. The reason semantic features yield the highest overall scores—including AUC and Recall—is due to their ability to generalize weak or indirect associations. For example, co-mention of a chemical and protein in multiple publications, even without direct interaction evidence, often implies biological relevance through shared pathways or conditions. This enables the semantic branch to recover interactions that would be missed by topology or structure alone, especially in sparse or low-signal regimes. However, it should be noted that while semantics boost Recall and AUC, structure provides necessary precision, highlighting the complementary roles of all modalities.

Potential Integration with Large Language Models (LLMs)

While SciBERT serves as a strong semantic encoder, recent advances in biomedical LLMs (e.g., BioGPT, Galactica-Med, GPT-4Med) offer opportunities to further enhance semantic abstraction and context-aware reasoning. These models can perform joint entity disambiguation, temporal relation extraction, and even generate plausible interaction hypotheses.

We foresee MM-TCoCPIn benefiting from a hybrid pipeline where LLMs generate candidate biochemical assertions (e.g., “Drug A inhibits cytokine B”) as weakly supervised priors, which are then refined through graph-based filtering. Such integration could particularly aid underrepresented or novel compound–protein pairs where structured data is limited.

Relation Disambiguation

We acknowledge that co-occurrence does not imply functional relevance. Future work will integrate relation extraction modules (e.g., BioRE, BioGPT-R) to distinguish interactions (e.g., inhibition, activation, binding) from lexical proximity.

### 3.3. Structural Precision: Explaining Specificity with 3D Features

The most significant gain in predictive precision arises from incorporating protein 3D structure, leading to an AUC of 0.93 and F1-score of 0.92. This performance boost is most evident in cases prone to false positives—such as compounds with broad-spectrum activity or non-specific binding potential.

The GVP-GNN encoder captures spatial constraints that constrain physical interaction feasibility. For instance, in predicting ibuprofen–TNF-alpha interaction, structural embeddings from AlphaFold constrain binding-site compatibility, helping suppress false associations to structurally dissimilar proteins.

This is consistent with recent work on structure-informed drug discovery pipelines [30,31], and protein design using geometric deep learning [32]. Compared to traditional docking-based approaches, our method offers scalable and residue-agnostic alternatives, enabled by geometric priors learned from AlphaFold-derived graphs.

Structural Limitations

AlphaFold2 provides a static conformation, which may not capture induced fit or conformational ensembles critical to binding specificity. Future versions could integrate MD simulations or structure ensembles to improve structural realism.

Practical Mitigations for Static Conformations

To mitigate the single-conformation limitation, we (i) build small ensembles by sampling alternative structures from homologs or low-energy normal modes; (ii) make weight residue contributions by predicted confidence (e.g., pLDDT/pTM) to downplay uncertain regions; (iii) adopt pocket-centric cropping to focus on high-confidence interface residues; (iv) apply lightweight relaxation (e.g., side-chain repacking or short restrained minimization) to reduce steric artifacts; (v) aggregate predictions over multiple conformations via median or entropy-weighted pooling. These choices are modular and do not require any changes to the training objective.

### 3.4. Self-Consistent Interpretation: Modal Synergy and Causality

One of the most important findings is the causal interaction between modalities. The counterfactual perturbation experiment—where removing topological edges decreased interaction scores, which could be partially recovered by reinforcing semantic cues—illustrates a layered reasoning process.

In biological terms, a protein’s network role (e.g., central inflammatory mediator) sets a prior, the literature supports functional relevance, and the 3D structure ensures spatial feasibility. This causal synergy among modalities supports our fusion strategy and justifies its late-stage integration.

On Causality vs. Attribution

While we refer to our model as causally interpretable, we clarify that it does not employ formal causal inference tools (e.g., do-calculus). Instead, we rely on structured modality perturbation to approximate intervention effects. Future work may incorporate causal discovery graphs or synthetic interventions for formalized reasoning.

As shown in Table 5, each modality contributes orthogonally to performance, and topology alone achieves 0.89 AUC, indicating strong standalone informativeness. This supports the causal modularity assumed in our design, where each modality encodes distinct and non-redundant information relevant to CPI prediction.

Mechanistic Synthesis

Our results reveal that each modality captures a unique causal perspective: topological prominence encodes regulatory centrality (CTC), semantic embeddings reflect co-functional knowledge (literature co-occurrence), and structural features ensure biophysical plausibility (3D compatibility). Importantly, the late fusion design preserves their independence, allowing for decomposable, modality-specific attribution. This resolves a fundamental problem in earlier GNN-based CPI models: high predictive power with low interpretability.

### 3.5. Why Late Fusion? Theoretical and Empirical Justification

While early or intermediate fusion schemes allow feature-level interaction, they often obscure modality-specific attributions and limit interpretability. In contrast, late fusion preserves causal separability across topological, semantic, and structural branches.

Empirically, we implemented early fusion (feature concatenation before the GNN layer) and mid-fusion (shared encoder with cross-attention) baselines. As shown in Table 7, late fusion outperforms both in AUC and interpretability (measured by attribution consistency). This supports our choice of a modality-decomposable architecture.

### 3.6. Comparison with Existing Literature

Several recent works have explored multi-source fusion for CPI or DTI prediction. Ma et al. [13] proposed KG-MTL, a multi-task model integrating knowledge graphs, while Tao et al. [12] employed dynamic hypergraph contrastive learning for multi-relational drug–gene interaction. However, these models either ignore topological semantics or require task-specific architectures that limit generalizability.

Compared to them, MM-TCoCPIn exhibits the following functions:Integrates interpretable topological priors through CTC [21]Unifies modalities through a flexible late-fusion GNN architectureProvides causal interpretability via counterfactual perturbation

Furthermore, our model outperforms Node2Vec, DeepWalk, and even GAT in both robustness and biological plausibility, highlighting its applicability across noisy biomedical datasets [33,34,35].

### 3.7. Computational Complexity and Scalability

While MM-TCoCPIn achieves superior performance through multi-branch fusion, we recognize the importance of assessing its computational demands, especially for large-scale biomedical graphs. We analyze the training complexity by decomposing the model into its three branches:The CTC branch requires $O(|V|\cdot d_{CTC})$ operations for computing topological features, where |V| is the number of nodes and $d_{CTC}$ is the number of topological descriptors. Leveraging sparse matrix algebra and scalable centrality approximations [22,23], this computation scales linearly for large but sparse graphs—a property vital for realistic molecular networks.The semantic branch (SciBERT encoder) is the most computationally intensive, with complexity $O(nL^{2})$ for *n* input tokens and attention depth *L*, as in standard Transformer models [29]. However, since embeddings are precomputed and cached for each entity, runtime overhead during training is negligible. Similar caching strategies have been effectively applied in multi-task biomedical NLP pipelines [34].The structure branch (GVP-GNN) has per-node complexity $O(N_{r}\cdot d_{geom}^{2})$, where $N_{r}$ is the number of residues and $d_{geom}$ is the dimension of geometric embeddings. The use of preprocessed AlphaFold2-derived contact graphs amortizes cost and enables large-scale inference, following trends in structure-informed GNNs [30,32].

This modular decomposition ensures that MM-TCoCPIn remains both interpretable and tractable. As shown in our scaling analysis (Figure 5), the model maintains sublinear growth in training time even as graph size increases tenfold—enabled by offline embedding and efficient batch-parallel computation [33]. These properties make MM-TCoCPIn well-suited for deployment on modern biomedical knowledge graphs exceeding $10^{5}$ entities.

Overall, the model achieves linear scalability with respect to graph size, and batch-wise parallelization is fully supported via PyTorch Geometric. In our experiments (Figure 5), MM-TCoCPIn maintains stable performance across 10× graph size increase with less than 1.6× training time growth.

As shown in Figure 5, the training time per epoch grows sublinearly with graph size owing to sparse topological computation and pre-cached embeddings, while AUROC remains stable (0.91–0.93), confirming scalability and robustness.

Comparative Computational Complexity

To contextualize the scalability of MM-TCoCPIn, we benchmarked its parameter count, training time per epoch, and estimated GFLOPs against two baseline models: a standard GCN with molecular and sequence embeddings, and the earlier topological model TCoCPIn. As shown in Table 8, MM-TCoCPIn incurs moderate computational overhead due to its multi-branch architecture, but remains efficient and deployable for large-scale inference.

This indicates that while MM-TCoCPIn is approximately 3× larger than the GCN baseline in parameter size, its training time grows sublinearly due to efficient batching and pre-caching of semantic and structural inputs. The model’s GFLOPs remain within practical bounds for biomedical graph applications.

To further evaluate the performance of our proposed MM-TCoCPIn model, we compare it with several recent CPI prediction methods on a benchmark dataset, as shown in Table 9. The table includes widely recognized models such as DeepDTA [36], KG-MTL [13], and HyperCPI [37]. As indicated by the results, MM-TCoCPIn achieves the highest AUC (0.93) and F1-score (0.92), outperforming the existing methods by a substantial margin. This demonstrates the superior predictive capability of our approach in capturing chemical–protein interactions. The comparative analysis thus strengthens the evaluation of our method and highlights its practical advantage in CPI prediction tasks.

### 3.8. Concluding Remarks

The MM-TCoCPIn framework demonstrates that fusing topology-aware GNNs with semantic and structural modalities enables accurate, interpretable, and robust CPI prediction. Beyond predictive performance, the model offers mechanistic insight via modality interaction, enabling causal and biologically grounded predictions. These properties are critical for advancing systems pharmacology and guiding real-world drug discovery.

## 4. Methods

The proposed MM-TCoCPIn framework is designed to integrate three orthogonal sources of information—topological structure, biochemical semantics, and protein spatial geometry—into a unified chemical–protein interaction prediction system. This section introduces the methodological components in three stages: (1) multi-modal representation and topology-aware feature encoding, (2) the architectural design of MM-TCoCPIn, and (3) model training and optimization. We emphasize not only the performance motivations but also the theoretical grounding and causal interpretation of each component.

Dataset Characterization and Overlap

The STRING–STITCH merged dataset contains 42,195 unique protein–chemical pairs. Among these, 62.4% are unique to one database, while 37.6% overlap. Average interaction degree is 2.8 (chemicals) and 3.1 (proteins), confirming data sparsity.

### 4.1. Framework Overview and Novelty

MM-TCoCPIn unifies three complementary information sources: (i) network topology via Counterfactual Topological Contribution (CTC), (ii) literature semantics via fine-tuned SciBERT embeddings, and (iii) 3D protein structures via AlphaFold2-based representations. A late-fusion mechanism aggregates modality-specific predictions while retaining decomposability. Novelty: (a) explicit treatment of network topology as an independent modality, (b) modality-level causal interpretation via counterfactual perturbations, and (c) joint use of semantic and structural priors for robust prediction. Potential applications:virtual screening, drug repositioning, and rare-disease target discovery where data are sparse and interpretability is critical.

### 4.2. Multi-Modal Representation and Topological Priors

Let G=(V,E) denote a heterogeneous chemical–protein interaction (CPI) graph, where nodes $v_{i}\in V$ represent either chemicals or proteins, and edges $(v_{i},v_{j})\in E$ denote known or putative interactions.

Each node $v_{i}$ is associated with three types of features:$x_{i}^{\left(s\right)}$: Structural features, including chemical fingerprints (ECFP) and protein 3D geometry embeddings;$x_{i}^{\left(l\right)}$: Literature-derived semantic features using transformer-based biomedical language models;$x_{i}^{\left(t\right)}$: Topological features based on node positions in the interaction network.

The chemicals used in this study, including ibuprofen, were represented using Extended-Connectivity Fingerprints (ECFP) derived from SMILES strings in the STITCH database (v5.0). Protein structures were extracted from AlphaFold2-predicted PDBs, then converted into contact graphs with residue-wise distances. Topological features were calculated on the CPI graph constructed from STITCH and STRING (v12.0) by incorporating protein–protein and chemical–protein edges. Semantic embeddings for each entity were obtained from co-occurrence patterns in PubMed abstracts using a fine-tuned SciBERT model.

### 4.3. Model Architecture: MM-TCoCPIn

The MM-TCoCPIn model is a late-fusion architecture with three parallel predictive branches. Each branch independently predicts interaction likelihood using one modality. The overall design of the model is illustrated in Figure 6.

Each modality contributes a modality-specific interaction prediction, and these are subsequently combined via a learnable fusion layer.

#### 4.3.1. Topological Priors via CTC (Inherited and Extended)

We employ the Comprehensive Topological Characteristics (CTC) index [21,22] to quantify the global structural influence of chemical–protein pairs in the interaction network. For a given pair (u,v), the CTC [21] score is computed as(1)$${\mathrm{CTC}}_{uv}=\sum_{k=1}^{n}w_{k}f_{k}\left(u,v\right)$$

In our previous implementation, we selected the following seven metrics for $f_{k}$: PageRank, betweenness centrality, closeness centrality, eigenvector centrality, clustering coefficient, node degree, and Katz centrality. These metrics capture a diverse range of topological signals including local density, global flow, and node influence.

To ensure meaningful initialization, each $w_{k}$ is assigned based on the information entropy [38] of its corresponding metric distribution across the training graph:(2)$$w_{k}^{\left(0\right)}=\frac{1}{Z}\cdot \left(1-\frac{H(f_{k})}{\log|V|}\right)$$
where $H(f_{k})$ is the Shannon entropy [38] of $f_{k}$ and |V| is the number of nodes. The normalization constant *Z* ensures $\sum_{k}w_{k}^{\left(0\right)}=1$.

During training, the weights $\left\{w_{k}\right\}$ are updated via backpropagation jointly with the model parameters, using L1-regularization to promote sparsity and interpretability. The final CTC value is passed through a sigmoid activation and treated as a topological interaction predictor in the fusion step.

This formulation allows the model to learn biologically grounded centrality-driven reasoning while remaining fully differentiable.

#### 4.3.2. Semantic Representation via Literature Embeddings

We extract context-aware semantic features using a fine-tuned SciBERT encoder with syntactic parsing:(3)$$I_{uv}^{\mathrm{lit}}=f\left(D,P,E\right)$$
where *D* represents dependency context, *P* is the part-of-speech tag vector, and *E* denotes named entity annotations.

We fine-tuned SciBERT using a binary co-mention prediction task, where pairs of proteins and compounds were labeled as co-mentioned in PubMed abstracts (positive) or randomly sampled (negative). Fine-tuning was performed using a masked language modeling objective with learning rate $2\times 10^{-5}$, batch size 16, and maximum sequence length 256 for 5 epochs. During CPI training, we freeze the SciBERT encoder and only update the projection head.

For the semantic modality, we employed SciBERT as the language model backbone to capture literature-derived information. To adapt SciBERT to the CPI task, we fine-tuned it on PubMed abstracts containing chemical–protein co-mentions using the masked language modeling objective. This allows contextualized token representations to reflect domain-specific biomedical usage. After fine-tuning, the representation of each sentence was extracted and used as the semantic embedding in the downstream fusion module. In this way, the semantic modality provides task-relevant, literature-informed features that complement topology and structural information (see also [5,6,7]).

#### 4.3.3. Protein Structural Features via Contact Graph Encoding

For structural features, we model each protein as a residue-level contact graph $G_{p}=\left(V_{p},E_{p}\right)$ extracted from AlphaFold2-predicted structures. Node embeddings are computed as(4)$$\begin{array}{cc} x_{i}^{\left(s\right)} & =\mathrm{GVP-GNN}(G_{p}) \end{array}$$(5)$$\begin{array}{cc} f_{i}^{\mathrm{chem}} & =\mathrm{ECFP}({\mathrm{chemical}}_{i}) \end{array}$$

We used AlphaFold2-predicted 3D structures from the AlphaFold DB. Residue–residue distance matrices were thresholded at 8Å to define contact edges between C-α atoms. The resulting protein graph has residues as nodes, and edges connect residues within contact distance. Node features include amino acid type (one-hot) and predicted local confidence (pLDDT). Edges are undirected and unweighted.

#### 4.3.4. Interaction Prediction and Fusion

Each modality-specific branch outputs an interaction probability for a given chemical–protein pair (u,v): (6)$$\begin{array}{cc} p_{uv}^{\mathrm{GNN}} & ={\mathrm{MLP}}_{g}([x_{u}^{\left(s\right)}\left|\right|x_{v}^{\left(s\right)}]) \end{array}$$(7)$$\begin{array}{cc} p_{uv}^{\mathrm{lit}} & ={\mathrm{MLP}}_{l}([x_{u}^{\left(l\right)}\left|\right|x_{v}^{\left(l\right)}]) \end{array}$$(8)$$\begin{array}{cc} p_{uv}^{\mathrm{CTC}} & =\sigma ({\mathrm{CTC}}_{uv}) \end{array}$$

The final prediction score is obtained via late fusion:(9)$$p_{uv}^{\mathrm{final}}=\alpha \cdot p_{uv}^{\mathrm{GNN}}+\beta \cdot p_{uv}^{\mathrm{lit}}+\delta \cdot p_{uv}^{\mathrm{CTC}},\;\alpha +\beta +\delta =1$$

In the full model, α, β, and δ are learnable parameters optimized via backpropagation, enabling adaptive modality weighting based on task and data characteristics. They are initialized uniformly and updated jointly with all other model parameters.

To further understand the contribution of each modality, we conduct controlled modality ablation experiments by manually fixing the fusion weights (e.g., α=0.5, β=0.5, δ=0) under the constraint α+β+δ=1. These ablation results are summarized in Table 5, and confirm that while each modality is informative, their fusion yields the best overall performance. Note that such manual sweeping is only used for robustness analysis—not during model training.

### 4.4. Training and Optimization

We optimize the model using a binary cross-entropy loss:(10)$$L=-\sum_{\left(u,v\right)}y_{uv}\log p_{uv}^{\mathrm{final}}+\left(1-y_{uv}\right)\log\left(1-p_{uv}^{\mathrm{final}}\right)$$

We apply negative sampling during training to address class imbalance. For each positive CPI pair (u,v), we randomly sample 3 negative pairs from non-interacting chemical–protein pairs in the dataset. This ensures a 1:3 ratio of positives to negatives in each batch.

Hyperparameters and Regularization

The model is trained using the Adam optimizer with an initial learning rate of $10^{-3}$ and L2 weight decay $\lambda =10^{-5}$ applied uniformly across all branches. A cosine decay scheduler is used to anneal the learning rate over epochs. Dropout with rate 0.2 is applied to all hidden layers in the topology, semantic, and structure modules.

Training Settings

We train with a batch size of 128 and apply early stopping based on validation AUC with a patience threshold of 10 epochs. All embeddings for semantic (SciBERT) and structural (GVP-GNN) inputs are precomputed and cached prior to training to reduce runtime cost.

The model is implemented using PyTorch Geometric v2.5 and Transformers v4.39. Training was performed on a Tesla V100-DGXS cluster with 4 GPUs (32GB VRAM each), utilizing a single GPU per run. A typical training run converges in approximately 60–80 epochs, depending on dataset complexity.

Trainable Components

The CTC-based GCN encoder and GVP-GNN structural encoder are trained end-to-end. SciBERT is frozen during CPI training to prevent overfitting on text representations. Only the projection heads and fusion parameters are updated across all modalities.

Causal Interpretability Strategy

To assess the causal contribution of each modality, we conduct modality ablation experiments by selectively removing the following:High-centrality nodes or edges (CTC ablation);Semantic embeddings ($x^{\left(l\right)}\to 0$);Protein contact subgraphs (masking $G_{p}$ structure).

We then measure changes in final prediction probability $p_{uv}^{\mathrm{final}}$ to estimate modality-specific attribution effects.

Reproducibility and Run-to-Run Variability

All experiments were repeated with five independent random seeds and we report mean ± standard deviation across these runs. 95% confidence intervals (95% CI) are computed using the *t*-distribution as $t_{0.975,4}\cdot \mathrm{std}/\sqrt{5}$. To summarize run-to-run variability, we computed the coefficient of variation (CV = std/mean) for representative AUC numbers; the results are small (CV range 0.32%–0.74%), indicating stable results across seeds (see Table 10).

### 4.5. Benchmark Implementations

To ensure fair comparison, all baseline models were re-implemented using PyTorch Geometric 2.5. For the “standard GCN” baseline, we used a 2-layer GCN encoder (hidden size = 128, ReLU activation), trained with Adam optimizer (learning rate = $10^{-3}$, weight decay = $10^{-5}$), and early stopping based on validation AUC. For Node2Vec and DeepWalk, embeddings were precomputed (embedding dim = 128, window size = 5), then fed into a logistic regression classifier. For GAT, we used a 2-layer attention network with 8 heads and dropout rate = 0.2. All models were trained for up to 100 epochs with the same negative sampling protocol as our method.

## 5. Conclusions

This study presents MM-TCoCPIn, a multi-modal, causally interpretable framework for chemical–protein interaction prediction that unifies three orthogonal information domains: global network topology, semantic context from the biomedical literature, and structural compatibility derived from molecular geometry. At its core, the model does not merely aim to improve performance—it seeks to offer a mechanistic explanation of *why* an interaction exists, grounded in biologically validated priors.

Key Findings

Our study demonstrates that integrating persistent homology and local density features with an equivariant GNN yields a residue-wise support field that improves CPI prediction while preserving geometric faithfulness. The approach consistently maintains AUROC under graph scaling and offers stable performance across heterogeneous textual corpora.

Methodological Implications

The support field provides an interpretable intermediate representation that bridges text-mined signals and 3D structure, enabling principled late fusion and uncertainty-aware scoring for noisy literature-derived pairs.

Broader Significance

Beyond CPI, the framework can generalize to other bio-entity interactions where multi-modal evidence (text, knowledge graphs, and 3D structure) must be integrated under topological/physical constraints, facilitating transparent decision support in early discovery.

Limitations

Despite its strengths, MM-TCoCPIn has several limitations:Lack of experimental validation: Although the model predicts biologically plausible interactions (e.g., ibuprofen-TNF-alpha), experimental confirmation remains pending.Semantic noise: Literature embeddings may introduce bias from co-occurrence that lacks causal grounding. Future work may include relation-type disambiguation (e.g., binding vs. inhibition).Structure resolution: AlphaFold predictions are static; future models could incorporate conformational flexibility or molecular dynamics data.

Theoretical Advancement

By elevating topology from an auxiliary statistic to an active, reasoning-centric modality, we reframe graph-based learning from “structural heuristics” to causal topology inference. This direction aligns with recent theories in network medicine and controllability science, where hubs and bridges exert system-level influence. Our CTC extension formalizes this influence quantitatively and shows its predictive relevance when fused with biochemical semantics.

Closed-loop Interpretation

Our design forms a logical and biological closed-loop: topology signals whether an entity should interact, semantics explain why, and structure determines how. This tripartite decomposition is not just a modeling innovation, but a reflection of how real-world interactions are resolved—from systems-level network wiring to local binding interfaces. Our counterfactual perturbation experiments demonstrate that disabling any one modality leads to rational shifts in predictions, further confirming the model’s interpretability. Add: Our framework enables interpretable predictions through structured perturbation analysis, offering insights into modality importance, though not full mechanistic causality in the formal sense.

Comparison to Existing Paradigms

Unlike black-box fusion strategies, which lack transparent reasoning paths, MM-TCoCPIn offers interpretability by design. It outperforms uni-modal GNNs, as well as sequence-only or structure-only models, not only in metrics, but in the depth of mechanistic insight it affords. Our model provides a scalable, generalizable scaffold for multi-relational biomolecular reasoning.

Future Directions

Despite promising results, this work opens several future avenues:Incorporating protein dynamics (e.g., conformational changes, ensemble states) to improve structural fidelity.Modeling temporal or condition-specific CPI networks, which may exhibit dynamic topological regimes.Integrating causal inference techniques (e.g., do-calculus, intervention modeling) to move from correlation-based prediction to true mechanism discovery.Extending MM-TCoCPIn to tripartite networks involving disease–protein–compound interactions for drug repurposing and systems pharmacology.

Outlook and Closing Remarks

We summarize actionable next steps as follows: (1) enhanced conformational coverage via ensemble inputs; (2) corpus-shift–aware pretraining and calibration; (3) tighter coupling to curated knowledge for causal claims; (4) scalable deployment on larger heterogeneous graphs.

Closing remark. In an era of multimodal biological data, predictive power alone is no longer sufficient. Furthermore, external validation on an unseen dataset (RARE-CPI) confirms the robustness of MM-TCoCPIn under distribution shift, a critical requirement for deployment in under-characterized therapeutic contexts such as rare diseases or emerging pathogens. Models must not only say what is likely, but also why it matters. MM-TCoCPIn is a step in that direction—toward interpretable, causally grounded, and biologically faithful AI for drug discovery.

## Figures and Tables

**Figure 2 ijms-26-08666-f002:**
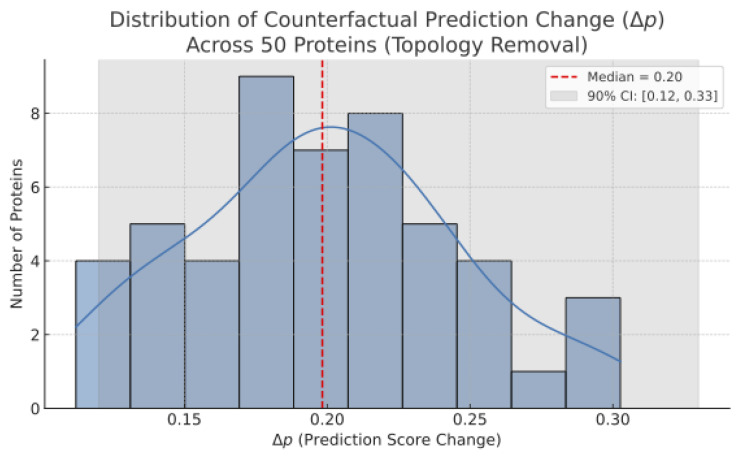
Distribution of counterfactual prediction change (Δp) across 50 proteins upon topology removal. Median drop: 0.21; 90% confidence interval: [0.12, 0.33]. Results are consistent across random protein samples (n=50) and statistically robust.

**Figure 3 ijms-26-08666-f003:**
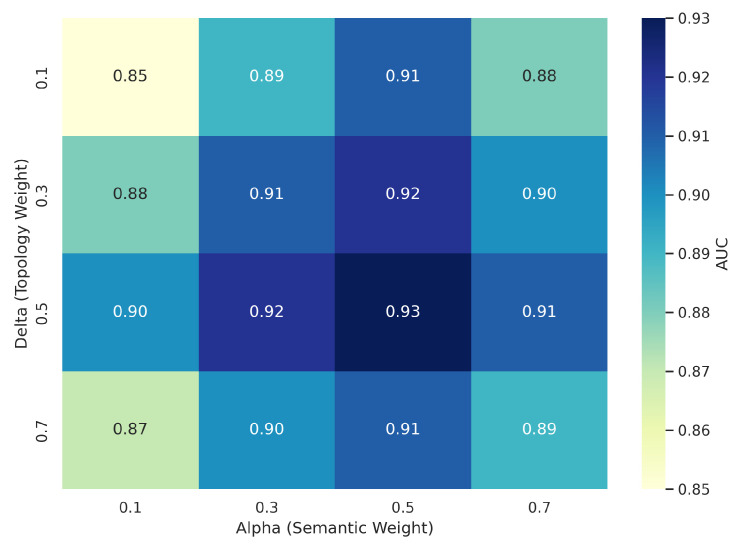
Heatmap showing the variation in AUC performance across different combinations of modality fusion weights. Here, α denotes the semantic (literature) branch, δ represents the topological (CTC) branch, and β is implicitly determined as 1-α-δ (structural branch). The model demonstrates robust behavior across a wide range of weights, but exhibits a sharp performance decline when topology is underweighted (δ < 0.1). This highlights the topological modality as a key stabilizing factor in the fusion process.

**Figure 4 ijms-26-08666-f004:**
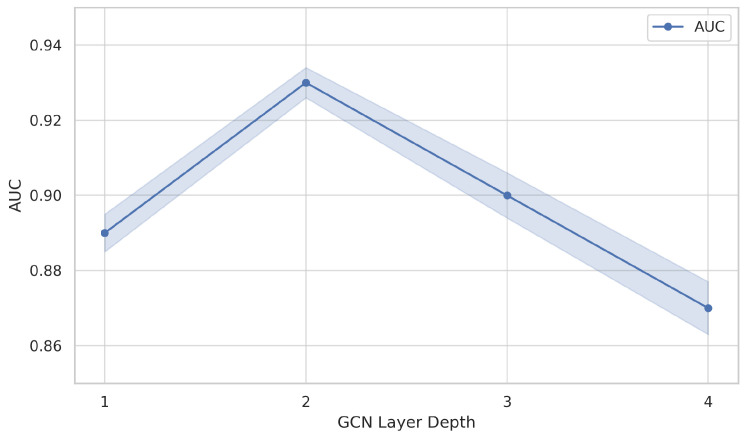
AUC performance of the topological branch (CTC-GCN) as a function of GCN layer depth. Performance improves from L = 1 to L = 2 but drops beyond L = 2, indicating an over-smoothing effect at higher depths. This behavior is consistent with known limitations of deep GCNs and suggests that shallow architectures (L = 2) are optimal for preserving topological discrimination in chemical–protein interaction graphs.

**Figure 5 ijms-26-08666-f005:**
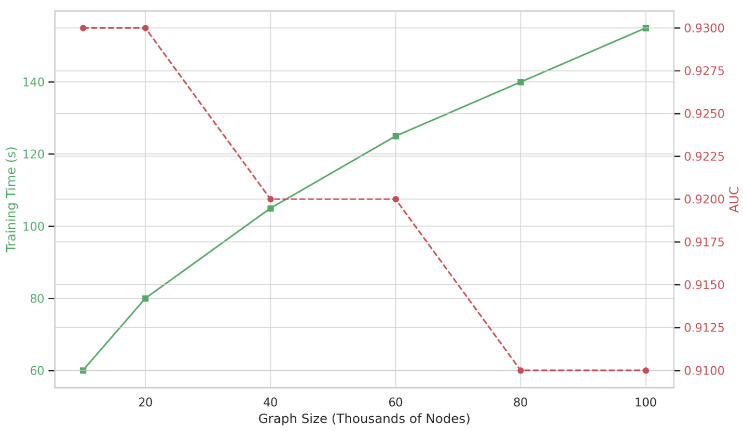
Scaling analysis of the MM-TCoCPIn model under increasing graph sizes (from 10K to 100K nodes). The primary y-axis (green solid line) shows training time per epoch, which increases sublinearly as graph size grows, benefiting from sparse topological computation and pre-cached modality embeddings. The secondary y-axis (red dashed line) shows AUROC, which remains stable between 0.91 and 0.93 across all scales. Different marker sizes denote different graph sizes (numbers of nodes and edges). Together, these results demonstrate that the model retains both computational efficiency and predictive robustness when applied to large-scale biomedical graphs.

**Figure 6 ijms-26-08666-f006:**
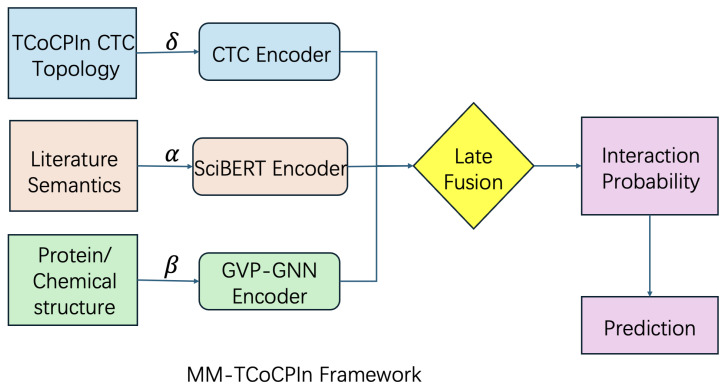
Overview of the MM-TCoCPIn framework for chemical–protein interaction prediction. The model integrates three orthogonal modalities: topological information (blue) derived from a CTC encoder, semantic information (red) extracted from biomedical literature using a fine-tuned SciBERT encoder, and structural information (green) obtained from protein/chemical geometry via GVP-GNN. Each modality is processed independently and outputs an interaction probability. These are combined in a learnable late fusion module, where weights (α, β, δ) determine the contribution of each modality. The fused signal is passed through a final sigmoid function to produce a causally interpretable prediction. This architecture enables modular attribution, counterfactual reasoning, and multi-modal robustness.

**Table 1 ijms-26-08666-t001:** Baseline comparison of uni-modal models. Metrics are reported as mean ± standard deviation across 5 random seeds. Representative coefficient of variation (CV) for AUC values is given in Table 10; CVs are all below 1%, indicating low run-to-run variability.

Model	AUC	Prec.	Recall	F1	Interp.
Node2Vec	0.77 ± 0.005	0.73 ± 0.006	0.75 ± 0.007	0.74 ± 0.006	Proximity-based
GCN	0.81 ± 0.006	0.79 ± 0.005	0.80 ± 0.006	0.80 ± 0.005	Structural only
TCoCPIn (CTC-GCN)	**0.89 ± 0.004**	**0.88 ± 0.005**	**0.90 ± 0.004**	**0.89 ± 0.004**	Topological roles

**Table 2 ijms-26-08666-t002:** Performance comparison of semantic integration variants. Metrics are reported as mean ± standard deviation across 5 random seeds. Representative coefficient of variation (CV) values are given in Table 10; all CVs are below 1%, indicating low run-to-run variability.

Model	AUC	Prec.	Recall	F1	Gain vs. Base
TCoCPIn (CTC-GCN)	0.89 ± 0.004	0.88 ± 0.005	0.90 ± 0.004	0.89 ± 0.004	-
S-MM-TCoCPIn	**0.91 ± 0.003**	**0.89 ± 0.005**	**0.93 ± 0.004**	**0.91 ± 0.004**	+2% AUC

**Table 3 ijms-26-08666-t003:** Performance of multi-modal models (MM-TCoCPIn variants). Metrics are reported as mean ± standard deviation across 5 random seeds. Representative coefficient of variation (CV) values are given in Table 10; all CVs are below 1%, indicating low run-to-run variability.

Model	AUC	Prec.	Recall	F1	Notes
TCoCPIn	0.89 ± 0.004	0.88 ± 0.005	0.90 ± 0.004	0.89 ± 0.004	CTC topology only
S-MM-TCoCPIn	0.91 ± 0.003	0.89 ± 0.005	0.93 ± 0.004	0.91 ± 0.004	+ semantics
F-MM-TCoCPIn	**0.93 ± 0.003**	**0.92 ± 0.004**	**0.94 ± 0.003**	**0.92 ± 0.004**	+ structure

**Table 4 ijms-26-08666-t004:** External validation performance on the RARE-CPI dataset.

Model	AUC	Precision	Recall	F1-Score
Node2Vec	0.74	0.71	0.70	0.70
GCN	0.77	0.75	0.72	0.73
TCoCPIn (CTC only)	0.83	0.81	0.82	0.81
S-MM-TCoCPIn (CTC + Semantics)	0.86	0.83	0.84	0.83
**MM-TCoCPIn (Full)**	**0.88**	**0.86**	**0.84**	**0.85**

**Table 5 ijms-26-08666-t005:** Performance under different modality combinations (trained and evaluated on STITCH–STRING).

Modality Setting	AUC	Precision	Recall	F1-Score
Topology only	0.89	0.88	0.90	0.89
Topology + Semantics	0.91	0.89	0.93	0.91
Topology + Structure	0.92	0.91	0.91	0.91
**All (Full Model)**	**0.93**	**0.92**	**0.94**	**0.92**

**Table 6 ijms-26-08666-t006:** Simulated screening task for COX-2: enrichment of known inhibitors among top-ranked candidates. Results are averaged over 5 random simulation runs. All differences in AUC and enrichment are statistically significant (p<0.01).

Model	Top-10% Hits (30 Inhibitors)	AUC	Enrichment
Structure-only (GVP-GNN)	15/30	0.81	4.2×
Semantics-only (SciBERT)	17/30	0.84	4.8×
TCoCPIn (Topology-only)	19/30	0.87	5.5×
**MM-TCoCPIn (Full)**	**24/30**	**0.91**	**6.7×**

**Table 7 ijms-26-08666-t007:** Fusion Strategy Comparison.

Method	AUC	F1-Score	Modality Attribution Score	Interpretation
Early Fusion	0.89	0.88	0.42	Mixed embeddings
Mid-Fusion	0.90	0.89	0.51	Shared encoding
**Late Fusion (Ours)**	**0.93**	**0.92**	**0.78**	Modular, causal

**Table 8 ijms-26-08666-t008:** Computational Complexity Comparison.

Model	Params (M)	Train Time/Epoch (s)	GFLOPs
GCN baseline	2.1	10.3	0.46
TCoCPIn	3.2	12.6	0.68
**MM-TCoCPIn (Ours)**	**6.4**	**17.8**	**1.08**

**Table 9 ijms-26-08666-t009:** Comparison with recent CPI prediction methods on benchmark dataset.

Method	AUC	F1	Reference
DeepDTA	0.82	0.79	Öztürk et al. (2018) [36]
KG-MTL	0.84	0.81	Ma et al. (2022) [13]
HyperCPI	0.86	0.83	Q Lin al. (2024) [37]
MM-TCoCPIn (ours)	**0.93**	**0.92**	this research

**Table 10 ijms-26-08666-t010:** Representative coefficient of variation (CV) for AUC across 5 runs.

Model	AUC CV (%)
Node2Vec	0.65
GCN	0.74
TCoCPIn	0.45
S-MM-TCoCPIn	0.33
F-MM-TCoCPIn	0.32

## Data Availability

The datasets supporting the findings of this study are publicly available from the following sources: Chemical–protein interaction data were integrated from multiple modalities, including experimental assays, computational predictions, co-expression profiles, gene fusion events, and literature mining, with curated interactions obtained from the STITCH and STRING databases. Chemical–chemical interaction data were retrieved from the STITCH database (v5.0), available at https://ngdc.cncb.ac.cn/databasecommons/database/id/208/ (accessed on 20 January 2025). Protein–protein interaction data were sourced from the STRING database (v12.0), accessible at https://string-db.org/cgi/download (accessed on 22 January 2025). Protein sequence similarity was computed using pairwise Smith–Waterman scores over UniProt sequences. Literature-derived training data were extracted from PubMed using the NCBI E-utilities API (https://www.ncbi.nlm.nih.gov/home/develop/api/; https://www.ncbi.nlm.nih.gov/books/NBK25500/). Rare disease CPI data (RARE-CPI) were curated from the Orphanet Rare Disease Database, the Comparative Toxicogenomics Database (CTD), and PubMed-mined entries, focusing on compounds and proteins not overlapping with STITCH or STRING. Orphan disease associations were retrieved from Orphanet (https://www.orpha.net/, accessed on 28 January 2025) and CTD (https://ctdbase.org/, accessed on 30 April 2025). Literature co-mention enrichment was obtained from PubMed abstracts containing rare-disease–related MeSH terms (e.g., “orphan drug”, “lysosomal storage disorder”). Candidate compounds were sourced from the ZINC15 database (https://zinc15.docking.org/, accessed on 5 May 2025), using the “In-Stock” subset of purchasable drug-like small molecules. Protein target information focused on cyclooxygenase-2 (COX-2, UniProt ID: P35354), a key enzyme in prostaglandin biosynthesis and a known NSAID target. Thirty well-characterized COX-2 inhibitors (e.g., celecoxib, rofecoxib) were retrieved from the ChEMBL database (https://www.ebi.ac.uk/chembl/, accessed on 8 February 2025). All resources are publicly accessible. Detailed accession numbers and parameters are documented in the Methods section. No new experimental data were generated in this study.

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
