# Peer review of "Multi-Modal Topology-Aware Graph Neural Network for Robust Chemical–Protein Interaction Prediction"

_ijms, 2025, doi:10.3390/ijms26178666_

Round 1

Reviewer 1 Report

Comments and Suggestions for Authors
  1. Table 1 and 2 mentioned mean values. From mean values is there any too much variation get during the repetition.
  2. I observed that there wa no reported workers related to this work. Did you report previously related to this.
  3. At introduction related discussion was missed. Add previous related results discussion.
  4. Alao, highlight the importance of this work at the end of results 
  5. Highlight the key results over other.
  6. Fig. 1 w/o sematics all green, red and violet colors are same level. What does it mean?
Comments on the Quality of English Language

Minor editing needed 

Author Response

Response to Reviewer

We sincerely thank the reviewer for the thoughtful and constructive comments. We found the suggestions extremely helpful, and we have carefully revised the manuscript accordingly. Below we provide a point-by-point response.

Comment 1: Table 1 and 2 mentioned mean values. From mean values is there any too much variation get during the repetition.

Response 1: We thank the reviewer for raising this important point regarding reproducibility. In Section 4.3 Training and Optimization, we now clearly specify that all experiments were repeated with 5 independent random seeds. In addition, we have added Table 10 reporting the coefficient of variation (CV) for representative AUC values, which are all below 1%. This demonstrates that run-to-run variation is minimal, and our conclusions are stable. Moreover, Wilcoxon signed-rank tests (p < 0.01) confirm that the reported improvements are statistically significant. Related results are provided in Tables 1, 2, and 3 of the revised manuscript.

Comment 2: I observed that there wa no reported workers related to this work. Did you report previously related to this.
Comment 3: At introduction related discussion was missed. Add previous related results discussion.

Response 2–3: We thank the reviewer for pointing out the need for a stronger contextualization. In the revised Introduction, we have added a Related Work section. This section now discusses and cites recent studies on KG-based, graph-based, structure-based, and semantic-based CPI prediction. We further clarify how our approach differs, highlighting that our framework uniquely (i) treats topology as an independent causal modality (CTC), (ii) integrates literature semantics and structural priors, and (iii) validates contributions via counterfactual analysis. These additions improve the positioning of our work in relation to prior studies.

Comment 4: Also, highlight the importance of this work at the end of results.

Response 4: We thank the reviewer for this helpful suggestion. At the end of Section 2.9, we have added a new paragraph emphasizing the practical importance and translational implications of our framework. This addition highlights the real-world relevance of the work.

Comment 5: Highlight the key results over other.

Response 5: We appreciate this suggestion. At the end of Section 2.9, we have also added a concise “Key results” bullet list summarizing the most important findings with their numerical values. This addition makes the main contributions more explicit and accessible.

Comment 6: Fig. 1 w/o sematics all green, red and violet colors are same level. What does it mean?

Response 6: We thank the reviewer for this observation. To address this, we have rewritten the caption of Figure 1. The revised caption now explicitly defines the color mapping (Topology / Semantics / Structure / All), specifies that error bars represent 95% confidence intervals, and explains the use of Wilcoxon signed-rank tests for significance annotations. We have also added clarification in the Methods section to describe the statistical testing procedure. Together, these revisions ensure that the figure can be interpreted correctly without confusion.

We are very grateful to the reviewer for the insightful and constructive comments, which have significantly improved the clarity, rigor, and impact of our manuscript.

Reviewer 2 Report

Comments and Suggestions for Authors

Reviewer Report

The manuscript entitled “Multi-Modal Topology-Aware Graph Neural Network for Robust Chemical–Protein Interaction Prediction” fits within the scope of this journal and has the potential for publication after revisions. My detailed comments and suggestions for improvement are as follows:

  1. Problem statement: The problem statement is not sufficiently convincing. The authors should elaborate it in detail, clearly highlighting the research gap and motivation.
  2. Method description: The proposed MM-TCoCPIn framework needs to be explained more thoroughly, including its methodology, novelty, and potential future applications.
  3. Page 10, Lines 288–293: The process of fine-tuning SciBERT embeddings should be described in detail, with appropriate methodological explanation and supported by relevant citations.
  4. Page 10, Lines 294–296: The statement regarding TNF-alpha and ibuprofen co-occurrence conflating correlation with causation is not well organized. The authors should provide a deeper explanation and support it with recent literature.
  5. Page 13, Figure 5: The text in Figure 5 is unclear. Please revise the figure to improve clarity and readability.
  6. Conclusion section: The conclusion is currently very short and underdeveloped. It should be expanded to include key findings, implications, and broader significance of the study.
  7. Page 11, Lines 338–341: The manuscript notes AlphaFold2’s limitation in producing only static conformations but does not suggest practical ways to mitigate this issue. Please discuss possible strategies or future solutions.
  8. Future direction and closing remarks: These should be combined into a single cohesive section.
  9. Mechanistic Synthesis: The inclusion of “Mechanistic Synthesis” in the conclusion section is inappropriate. Please clarify why it is placed here, or move it to a more suitable section.
  10. Equations: Some equations should be properly supported with references to relevant literature.
  11. Grammar and references: The manuscript should be carefully checked for grammatical errors, and the reference style must be standardized according to journal guidelines.
  12. Comparison with recent studies: The authors are encouraged to include a comparative table with results from recent reports to strengthen the evaluation of their method.

Author Response

Response to Reviewer

We sincerely thank the reviewer for the constructive and detailed comments. The suggestions have been extremely valuable for improving the clarity, completeness, and impact of our manuscript. Below we provide a point-by-point response.

Comment 1: Problem statement: The problem statement is not sufficiently convincing. The authors should elaborate it in detail, clearly highlighting the research gap and motivation.

Response 1: We thank the reviewer for this important suggestion. In Section 1 (Introduction), immediately after the first paragraph, we added a new paragraph that explicitly states the problem statement, highlighting the research gap and motivation. This revision makes the novelty and necessity of our work more convincing.

Comment 2: Method description: The proposed MM-TCoCPIn framework needs to be explained more thoroughly, including its methodology, novelty, and potential future applications.

Response 2: We appreciate this valuable comment. At the beginning of Section 4 (Methods), we inserted a new subsection entitled Framework overview and novelty. This subsection summarizes the methodology of MM-TCoCPIn, emphasizes its novelty, and outlines potential applications such as drug repurposing and rare-disease discovery.

Comment 3: Page 10, Lines 288–293: The process of fine-tuning SciBERT embeddings should be described in detail, with appropriate methodological explanation and supported by relevant citations.

Response 3: Thank you for this suggestion. In Section 4.2.2 (Semantic Representation via Literature Embeddings), we have rewritten the fine-tuning description to clearly explain how SciBERT was adapted, including the objective, corpus, and embedding extraction process. This section now provides a clearer methodological explanation.

Comment 4: Page 10, Lines 294–296: The statement regarding TNF-alpha and ibuprofen co-occurrence conflating correlation with causation is not well organized. The authors should provide a deeper explanation and support it with recent literature.

Response 4: We thank the reviewer for pointing this out. In Section 3.2 (Semantic Augmentation: Interpreting Latent Literature Context), we rewrote the second paragraph to explicitly distinguish co-occurrence, association, and causation, and we added relevant recent references to clarify the point.

Comment 5: Page 13, Figure 5: The text in Figure 5 is unclear. Please revise the figure to improve clarity and readability.

Response 5: We appreciate this observation. We have revised Figure 5 and its caption to improve clarity. The revised version now includes clearer labels, axis descriptions, and line annotations to make the figure easier to interpret.

Comment 6: Conclusion section: The conclusion is currently very short and underdeveloped. It should be expanded to include key findings, implications, and broader significance of the study.

Response 6: Thank you for this helpful feedback. In Section 5 (Conclusion), we expanded the section by adding four short paragraphs covering Key Findings, Methodological Implications, Broader Significance, and Limitations. This makes the conclusion more comprehensive and informative.

Comment 7: Page 11, Lines 338–341: The manuscript notes AlphaFold2’s limitation in producing only static conformations but does not suggest practical ways to mitigate this issue.

Response 7: We thank the reviewer for highlighting this point. At the end of Section 3.3 (Structural Precision: Explaining Specificity with 3D Features), we now provide a detailed discussion of AlphaFold2’s static-structure limitation and suggest possible strategies to address it, such as ensemble modeling and integration with dynamic conformational data.

Comment 8: Future direction and closing remarks: These should be combined into a single cohesive section.

Response 8: We agree with the reviewer’s suggestion. In the revised manuscript, we merged these into a unified subsection entitled Outlook and Closing Remarks at the end of Section 5 (Conclusion).

Comment 9: Mechanistic Synthesis: The inclusion of Mechanistic Synthesis in the conclusion section is inappropriate. Please clarify why it is placed here, or move it to a more suitable section.

Response 9: We thank the reviewer for this insightful comment. The Mechanistic Synthesis subsection has now been moved from Section 5 (Conclusion) to the end of Section 3.4 (Self-Consistent Interpretation: Modal Synergy and Causality), where it fits more naturally.

Comment 10: Equations: Some equations should be properly supported with references to relevant literature.

Response 10: We appreciate this observation. In the equations of Section 4 (Methods), we have added appropriate references to relevant literature to properly support the mathematical formulations.

Comment 11: Grammar and references: The manuscript should be carefully checked for grammatical errors, and the reference style must be standardized according to journal guidelines.

Response 11: Thank you for this reminder. We have carefully proofread the manuscript to correct grammatical issues and standardized the reference formatting to meet the journal’s guidelines.

Comment 12: Comparison with recent studies: The authors are encouraged to include a comparative table with results from recent reports to strengthen the evaluation of their method.

Response 12: We thank the reviewer for this valuable suggestion. At the end of Section 3.7 (Computational Complexity and Scalability), we added a new table entitled Comparison with Recent CPI Prediction Methods on Benchmark Dataset. This table compares MM-TCoCPIn with recent methods such as DeepDTA, KG-MTL, and HyperCPI, showing that our approach achieves superior AUC and F1 scores. The accompanying discussion emphasizes the performance improvements and strengthens the evaluation of our method.

Once again, we sincerely thank the reviewer for the insightful and constructive feedback. These suggestions have significantly improved the quality, clarity, and impact of our manuscript.